# Emulsifying Properties of Rhamnolipids and Their In Vitro Antifungal Activity against Plant Pathogenic Fungi

**DOI:** 10.3390/molecules27227746

**Published:** 2022-11-10

**Authors:** Dongmei Li, Weiyi Tao, Dinghua Yu, Shuang Li

**Affiliations:** 1College of Biotechnology and Pharmaceutical Engineering, Nanjing Tech University, Nanjing 211816, China; 2College of Food Science and Light Industry, Nanjing Tech University, Nanjing 211816, China

**Keywords:** rhamnolipid, biosurfactant, emulsification, pesticide adjuvants, antifungal activity

## Abstract

**Simple Summary:**

The components of the rhamnolipid products of various sources are usually significantly different. This may hinder the application of rhamnolipid products. The rhamnolipid products with known structural components were used to test their emulsifying properties toward solvents in pesticide and their antifungal performance against plant pathogenic fungi. The results showed that the different components of rhamnolipids had no significant effect on the emulsifying performance, however, increased amounts of di-rhamnolipids in products exhibited stronger antifungal activity. We hope this work will be helpful to promote the application of rhamnolipids as pesticide adjuvants.

**Abstract:**

Rhamnolipids have significant emulsifying activity and the potential to become a component of pesticide emulsifier. Rhamnolipids are usually composed of two main components: mono-rhamnolipids (Rha-C_10_-C_10_) and di-rhamnolipids (Rha_2_-C_10_-C_10_). The proportion of di-rhamnolipids in the products ranged between 15% and 90%, affected by the production strains and fermentation process. In this paper, three kinds of rhamnolipid products containing di-rhamnolipids proportions, of 25.45, 46.46 and 89.52%, were used to test their emulsifying ability toward three conventional solvents used in pesticide (S-200, xylene, cyclohexanone) and antifungal activities against five strains of plant pathogenic fungi (*Phytophthora capsici*, *Phytophthora parasitica var.nicotianae*, *Colletotrichum destructivum*, *Colletotrichum sublineolum*, *Fusarium oxysporum*). The results indicated that although the CMC of the three rhamnolipids were significantly different, their emulsification properties had no remarkable differences, at a concentration of 10 g/L. However, their antifungal activities were significantly different: the more di-rhamnolipids, the stronger the antifungal activity. This work helps to promote the application of rhamnolipids as pesticides adjuvants.

## 1. Introduction

Rhamnolipids are one of the most widely studied biosurfactants. within comparison to chemical surfactants, rhamnolipids are highly biocompatible and biodegradable, and have potential applications in agriculture, cosmetics, food, microbial enhanced oil recovery (MEOR), environmental remediation and other industrial fields [1,2,3,4,5]. Rhamnolipids are metabolites of microorganisms that often exist in the form of mixtures, consisting of one or two units of rhamnose attached to one or two fatty acid chains with C_8_–C_14_ carbon atoms, which may or may not be saturated, primarily including Rha-C_10_-C_10_, Rha_2_-C_10_-C_10_ and Rha-C_10_ [6,7]. The specific composition of rhamnolipids is closely related to the production strain, fermentation substrate and process conditions, and the proportion of di-rhamnolipids are usually between 15–90% [8,9,10].

The composition of rhamnolipids have a great influence on their physical and chemical properties and application effects. Wu et al. [11] compared the surface/interface properties and aggregation behavior of mono-rhamnolipids and di-rhamnolipids, and found that the surface activity of mono-rhamnolipids was superior that of di-rhamnolipids. Rocha et al. [8] compared mono-rhamnolipids and di-rhamnolipids in MEOR applications and found that all rhamnolipids homologues had potential for MEOR applications. However, mono-rhamnolipids stood out due to their robustness under various physical and chemical conditions, which resemble an oil production reservoir. In terms of biological inhibition, Stanghellini and Miller [12] used methods such as column chromatography to separate mono-rhamnolipids and di-rhamnolipids, and found that di-rhamnolipids were comparable or better than mono-rhamnolipids in causing cleavage. However, rhamnolipids were usually produced and used as a mixture of homologs, due to the uneconomical separation and extraction process in practical applications [13]. As the homolog ratios of rhamnolipids products may affect their performance, we found it necessary to compare the emulsification capability and antifungal activity of rhamnolipids with varying contents of mono-rhamnolipids and di-rhamnolipids.

## 2. Materials and Methods

### 2.1. Production, Purification and Characterization of Rhamnolipid Products

Three rhamnolipid products were obtained by fermentation with *Pseudomonas aeruginosa* YM4, *Pseudomonas aeruginosa* HN and *Pseudomonas aeruginosa* PAO1, respectively. Strain *Pseudomonas aeruginosa* YM4 could efficiently produce di-rhamnolipid as the predominant component with glycerol as its carbon source [10]. Strain HN was isolated from oil contaminated soil; and strain PAO1 was kindly given by Prof. Xiaoyu Yong (@njtech).

For seed culture, all the strains were cultured in Luria-Bertani (LB) medium (10 g/L peptone, 5 g/L yeast extract, 10 g/L NaCl) at 200 rpm and 37 °C for 12 h, respectively. Next, it was transferred into the 250 mL baffled shake flask, containing 50 mL fermentation medium, at a 3% (v/v) inoculation rate. The fermentation for rhamnolipid production was continued at 37 °C and 180 rpm for 96 h. The fermentation medium comprised (g/L) NaNO_3_ (8), K_2_HPO_4_·3H_2_O (4), KH_2_PO_4_ (4), MgSO_4_·7H_2_O (0.2), CaCl_2_ (0.1), yeast extract (1), trace elements solution (2.5 mL/L), and carbon sources (30), respectively. The trace elements solution contained (g/L) FeCl_3_ (0.16), CuSO_4_ (0.15), ZnSO_4_·7H_2_O (1.5), and MnSO_4_·H_2_O (1.5). For strain YM4, glycerol was used as the carbon source; for the other two strains, soybean oil was used as the carbon source. After fermentation, the cell free supernatant was collected for rhamnolipids extraction.

As described previously [10], rhamnolipids were precipitated by acid and extracted by ethyl acetate extraction; finally, the rhamnolipid products were obtained by rotary evaporation. As described previously [10], rhamnolipids were detected directly, without derivatization, by HPLC using an Evaporative Light Scattering Detector (ELSD).

### 2.2. Materials and Fungi

A hydrocarbon oil (S-200), which is environmentally friendly, containing aromatic hydrocarbons with high-flash points (104 °C) and low toxicity, with a density of 0.996 g/mL, was obtained from EXXON. The analytical grade xylene and cyclohexanone were supplied by Shanghai Zhanyun Chemical Co., Ltd., Shanghai. Lingfeng Chemical Reagent Co., Ltd. Sodium dodecyl sulfate (SDS) was acquired from Sinopharm Chemical Reagent Co., Ltd. Technical-grade polyoxyethylene castor oil ethers (EL-40) were supplied by Shangdong Usolf Chemical Technology Co., Ltd.

Phytophthora capsici, Phytophthora parasitica var.nicotianae, Colletotrichum destructivum, Colletotrichum sublineolum, and Fusarium oxysporum were obtained as gifts from Professor Ling Qing (College of Plant Protection, Southwest University, Chongqing, China). Fungal cultures were maintained on Potato Dextrose Agar (PDA) at 28 °C.

### 2.3. Determination of the Critical Micelle Concentration (CMC) of Rhamnolipids

We determined the critical micelle concentration by measuring surface tension profiles of rhamnolipids dilutions [14]. We measured the surface tension changes with different concentrations of rhamnolipid solutions (0–200 mg/L) using an automatic tensiometer (BZY-3B, Shanghai Automation Instrumentation Sales Center, Shanghai, China) at room temperature.

### 2.4. Determination of the Emulsification

The surfactant solution was prepared at 10 g/L. 1.5 mL surfactants and 0.5 mL solvents (S-200, xylene, cyclohexanone) were mixed in an emulsifying vial at different pH. The vials were vortexed at high speed for 5 min, and subsequently kept still at room temperature for 7 days.

### 2.5. In vitro Bioassay for Antifungal Activity

#### 2.5.1. Evaluation of Suppression of Mycelial Growth

The antifungal ability of rhamnolipids was tested in vitro by following the method of Hultberg et al. [15]. Different concentrations (0, 5, 25, 50, 100, and 200 mg/L *w*/*v*) of rhamnolipids were separately mixed with PDA, and then sterilized and poured into 90 mm Petri dishes. Freshly growing mycelia (7 mm each) were transferred to the middle of the plates. The plates were then incubated at 28 °C until the control treatment (0 mg/L) covered the entire plate. The diameters of the mycelium growth were determined and compared with the control, and then the inhibition rates were calculated.

#### 2.5.2. Antifungal Activity against Fungal Spores

The refrigerated fungal mycelium was activated on a PDA plate for 6 days. The PDA colony covered with mycelia was punched with a sterilized puncher. Five pieces were picked and inoculated in the potato dextrose broth (PDB; Difco), and incubated at 28 °C, 120 r/min for 4 days. The culture was filtered through four layers of gauze to obtain spore suspension and the spore’s concentration was adjusted to 10^7^ spore/mL with a hemocytometer, under optical microscopy.

The antifungal activities of various concentrations of rhamnolipid products were tested against *F. oxysporum*, as outlined by Kim et al. [16], with some modifications. The rhamnolipids concentrations were 0, 5, 25, 50, 100, and 200 mg/L(*w*/*v*) in PDB. Spores suspension (0.2 mL) was mixed with 4 mL PDB. To compare the effect of rhamnolipids on spore germination, a positive control (only PDB) and a negative control (PDB + fungal spore suspension) were kept. We set up three repeats, and all were incubated at 28 °C. The number of germinated spores was counted on a hemocytometer, under optical microscopy.

### 2.6. Data Analysis

The data represent the arithmetical averages of three replicates and the error bars indicate the standard deviations. The antifungal activities of rhamnolipids at different concentrations were analyzed using the one-way analysis of variance (ANOVA), followed by least significant differences (LSD) test (*p* < 0.05). Statistical analysis was performed using the social science statistical package Origin 2020 with the plugin Paired Comparison Plot installed.

## 3. Results

### 3.1. Proportions of Rhamnolipid Products

The HPLC profiles of the rhamnolipid products are shown in Figure 1. It is evident that there were only two main components in the products: di-rhamnolipids (Rha_2_-C_10_-C_10_) (10.577 min) and mono-rhamnolipids (Rha-C_10_-C_10_) (13.375 min) [10]. The peak area ratio was the proportion of the rhamnolipid structure in the homologues. The detailed homology ratios of the three rhamnolipid products are shown in Table 1. The di-rhamnolipids contents of the three rhamnolipid products were approximately 25.45, 46.46 and 89.52%, and then the three products were named DR_25_, DR_46_ and DR_90_, respectively.

### 3.2. The Critical Micelle Concentration (CMC)

As shown in Figure 2, all three of the rhamnolipids could significantly reduce the surface tension of water, from 72.0 to 28.5 mN/m. The CMC values of DR_25_, DR_46_, and DR_90_ were 57.2, 65.6 and 75.2 mg/L, respectively.

### 3.3. Emulsifying Activity of Rhamnolipids on Solvents in Pesticides

S-200, xylene and cyclohexanone are three typical organic solvents commonly used in pesticides, and were selected as the organic phase to test the emulsifying activity of the three rhamnolipid products. The main component of S-200 is alkane; xylene is an aromatic hydrocarbon, which is a mixture of industrial grade *o*-xylene, *m*-xylene and *p*-xylene; cyclohexanone is one of the most polar substances in organic solvents.

#### 3.3.1. Emulsification of Chemical Emulsifiers and Rhamnolipids

SDS is a chemical emulsifier commonly used in the laboratory and EL-40 is a chemical emulsifier commonly used in pesticide formulations. All of the emulsifier solutions were used at the concentration of 10 g/L, pH 7.0. The results (Table 2) showed that the preferred solvent of SDS was cyclohexanone. On the contrary, the emulsification of EL-40 to cyclohexanone was poor, and xylene and S-200 were suitable solvents for EL-40. The emulsifying activities of rhamnolipids were similar to EL-40. Rhamnolipids had a good emulsifying effect on alkanes.

#### 3.3.2. The Effect of pH on Emulsification

The emulsifying activities of the rhamnolipids solution were determined under acidic (pH 5.5), neutral (pH 7.0) and alkaline (pH 8.5) conditions, respectively. As shown in Table 3, no emulsification layer formed in the cyclohexanone solvents. For S-200 and xylene, the emulsification layer could be formed, but they were unstable and easy to demulsify under acidic conditions. With respect to the same pH, there were no significant differences in the emulsification layer between each sample. Thus, the optimal pH for rhamnolipid solution was alkaline (pH 8.5).

#### 3.3.3. Emulsification Layer and Emulsified Particles Formed by Rhamnolipid Products

The rhamnolipids solution was used at the concentration of 10 g/L, pH 8.5. The emulsification layer and emulsified particles formed by the three rhamnolipids are shown in Figure 3. Both S-200 and xylene could form a stable emulsification layer, both with the same height. Furthermore, the morphology of the emulsified particles was observed under the microscope. The emulsified particles of xylene were larger than those of S-200, meaning that the emulsification layer of S-200 was more stable, and S-200 was the optimal solvent for rhamnolipids. With respect to the same solvent, there were no significant differences in the sizes of the emulsified particles.

### 3.4. Evaluation of Antifungal Activity of Rhamnolipid in Vitro

#### 3.4.1. Suppression of Mycelial Growth

The rhamnolipid products were used to investigate their mycelium inhibitions against five strains of plant pathogenic fungi: two strains of *Phytophthora* (*P. capsici, P. nicotianae*), two strains of *Colletotrichum* (*C. sublineolum, C. destructivum*) and one strain of *Fusarium* (*F. oxysporum*).

The average mycelium inhibition rates of the three rhamnolipid products (200 mg/L) against the five plant pathogenic fungi are shown in Table 4: for *P. capsici*, the mycelium inhibition rate of DR_90_ could reach 100%, while the mycelium inhibition rates of DR_25_ and DR_46_ were only 62.89 and 71.22%, respectively. For *P. nicotianae*, the mycelium inhibition rate of DR_90_ could reach 66.21%, while that of DR_25_ and DR_46_ were only 29.10 and 34.26%, respectively. For the two strains of *Colletotrichum* and one strain of *Fusarium*, the mycelial inhibition rates of DR_90_ were the best, followed by DR_46_, and DR_25_ were the worst.

The antifungal effects of rhamnolipids are shown in Figure 4. Both the concentration and the contents of rhamnolipids were key factors affecting their fungistatic activity. The more di-rhamnolipids, the stronger the antifungal activity was. For the two *Phytophthora* strains, the DR_90_ exhibited better activity against hyphal growth than the other two samples, at all concentrations. Both the DR_90_ and DR_46_ samples showed better antifungal activity than DR_25_ against the two *Colletotrichum* strains. The overall results showed that for the same strain, the mycelial inhibition rate of DR_90_ was the highest, and that of DR_25_ was the lowest, meaning the di-rhamnolipids component had a distinct advantage.

#### 3.4.2. Suppression of Spore Germination

Spores of *F. oxysporum* were used as the subject. Acting on the spore germination, when the concentration of rhamnolipids increased, the effects of inhibition were enhanced. However, there was no significant difference between the effects of the three rhamnolipids, and the inhibition rates were all approximately 50% at the concentration of 200 mg/L (Figure 5).

## 4. Discussion

Rhamnolipids often exist in the form of homologs, and different components have different properties [11]. Previously, the rhamnolipids were separated and purified to test which component worked best in different cases. The mixed products have to be used due to the high cost of separating homologs. Therefore, it is more valuable to explore the influence of the homolog ratios on their application.

In fact, as a highly active biosurfactant, rhamnolipids demonstrate a good emulsifying activity [17,18], and if rhamnolipids can be used to replace part of the chemical emulsifier used in pesticide, it will help to improve the environmental friendliness of the products. However, the influence of homolog ratios on emulsifying activity is still unclear. The purpose of this work was to promote the application of rhamnolipids as the formulation component of pesticide, by studying the effect of the components in rhamnolipids on their emulsifying activity toward the solvents widely used in pesticide. Rhamnolipids are an anionic surfactant. Lovaglio et al. [19] demonstrated that the emulsifying activity of rhamnolipid was affected by pH. Onaizi et al. [20] also reported that rhamnolipids had pH-responsive behavior to petroleum emulsification, with acid demulsification and alkaline re-emulsification. In this work, we reached the same conclusion that the optimum pH for emulsification activity in rhamnolipid was above seven. In addition, the HLB (hydrophile–lipophile balance) values can be used as a rough guide for surfactant selection [21]. The HLB values of the DR_25_, DR_46_ and DR_90_ were all around 12 (data not shown), and were close to that of EL-40 (HLB = 13.6).It is notable that the emulsification property of the rhamnolipids were similar to that of EL-40, and they all exhibited good emulsification ability toward alkane and aromatic hydrocarbon solvents; this suggests that rhamnolipids should be a potential emulsifier used in pesticide.

Previous studies have reported differences in emulsification between mono-rhamnolipids and di-rhamnolipids [17,22], but the working concentrations of rhamnolipids were very low, at approximately 100 mg/L or around CMC values. The results widely agree that homolog ratios in rhamnolipids have an important impact on performance. In pesticide formulas, emulsifiers are the key functional substance in the preparation of stable water emulsions, with an approximate dosage of 2–8% [23,24]. In this work, it has been proven that the emulsification capability had no significant difference among the rhamnolipids, with various homolog ratios at the concentration of 10 g/L. This suggests that homolog ratios of rhamnolipids have no impact on the emulsifying function in pesticide. This result helps to promote the application of diverse rhamnolipids.

Chemical drugs are the main functional substances used in the control of agricultural diseases, however, they also cause serious environmental pollution. Drug residues on crops also cause damage to the human body [25]. Subsequently, low-toxicity or non-toxic biological control methods have gradually attracted attention [26,27]. In terms of biology, rhamnolipids have a good inhibitory effect on phytopathogenic fungi, and therefore have the potential to become biological fungicides [28,29,30]. In 2016, Soltani et al. [31] separated mono-rhamnolipids (Rha-C_10_-C_10_) and di-rhamnolipids (Rha_2_-C_10_-C_10_) components and compared their ability to stop the zoospores among mono-rhamnolipids, di-rhamnolipids and the mixture. They found that the rhamnolipids mixture worked best, with a minimum concentration of 8–10 mg/L. Subsequently, increasing reports have suggested that the rhamnolipid mixture had an inhibitory ability on the mycelium growth and on the germination of the spores of various pathogens, however, the effects were significantly different. The inhibition rate of spores and mycelium growth on *Colletotrichum capsica* were 59.43 and 48.27% at 100 mg/L [28]. The inhibition rate of spores and mycelium growth on *Colletotrichum falcatum* were 86.63 and 67.8% at 100 mg/L [29]. For *Fusarium verticillioides FS7* [32], the mycelium growth inhibition rate was 82.23% at 200 mg/L. However, previous work has only suggested that the rhamnolipids had an inhibitory effect on the pathogen and did not state which had a better inhibitory effect by comparing the rhamnolipids samples with the standards. In fact, the effect of the rhamnolipid components on their biological activity remains unclear. In the present study, rhamnolipid products with known homolog ratios were used to act against five strains of plant pathogenic fungi, and we found that the rhamnolipids containing more di-rhamnolipids had better inhibitory activity on mycelium growth. This result helps to tailor the rhamnolipid products suitable for the inhibition of pathogenic fungi.

Overall, rhamnolipids had great potential application in pesticide formula. On one hand, the emulsifying properties of rhamnolipids can be used as emulsifiers; and on the other hand, the fungistatic activity of rhamnolipids can be used as fungicides.

## 5. Conclusions

The mono-rhamnolipids (Rha-C_10_-C_10_) and di-rhamnolipids (Rha_2_-C_10_-C_10_) components were the two main components in rhamnolipids. Rhamnolipids exhibited excellent emulsification and antifungal properties in this work. Although the physicochemical properties of di-rhamnolipids and mono-rhamnolipids were different, the emulsification capability of rhamnolipids with various homolog ratios had little difference at a high concentration (10 g/L). However, the biological activity of rhamnolipids were significantly affected by homolog ratios, and the di-rhamnolipids exhibited higher activity in inhibiting mycelium growth. Rhamnolipids with a greater di-rhamnolipids content were the best choice for fungal inhibition.

## Figures and Tables

**Figure 1 molecules-27-07746-f001:**
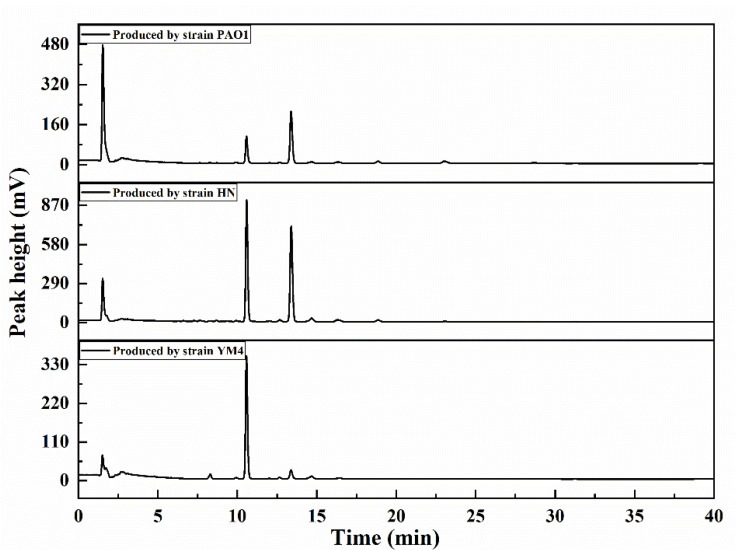
HPLC characterization of three rhamnolipid products.

**Figure 2 molecules-27-07746-f002:**
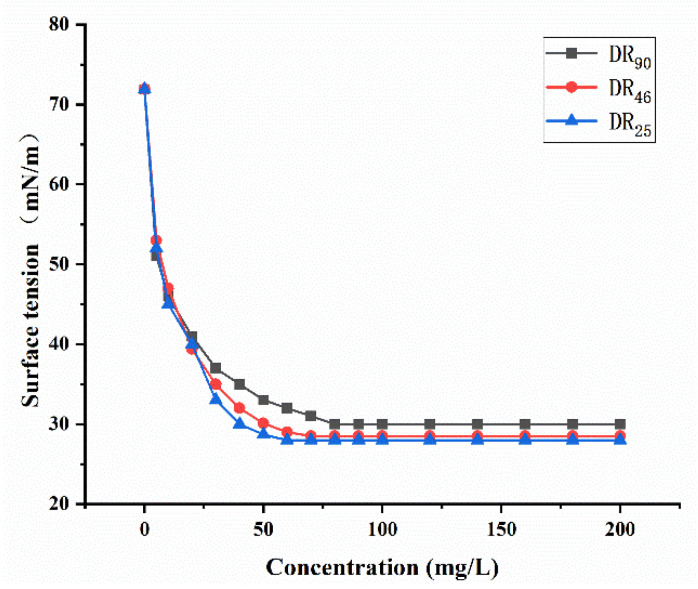
The CMC determination for rhamnolipids.

**Figure 3 molecules-27-07746-f003:**
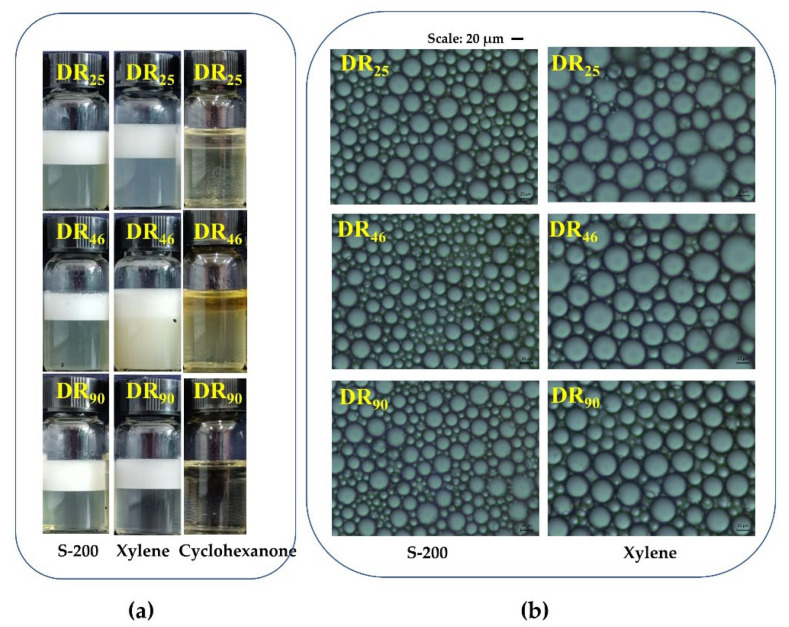
Emulsifying activity of DR_25_, DR_46_ and DR_90_ toward S-200, xylene and cyclohexanone. (**a**) Emulsification layer formed in solvents; (**b**) The morphology of emulsified particles.

**Figure 4 molecules-27-07746-f004:**
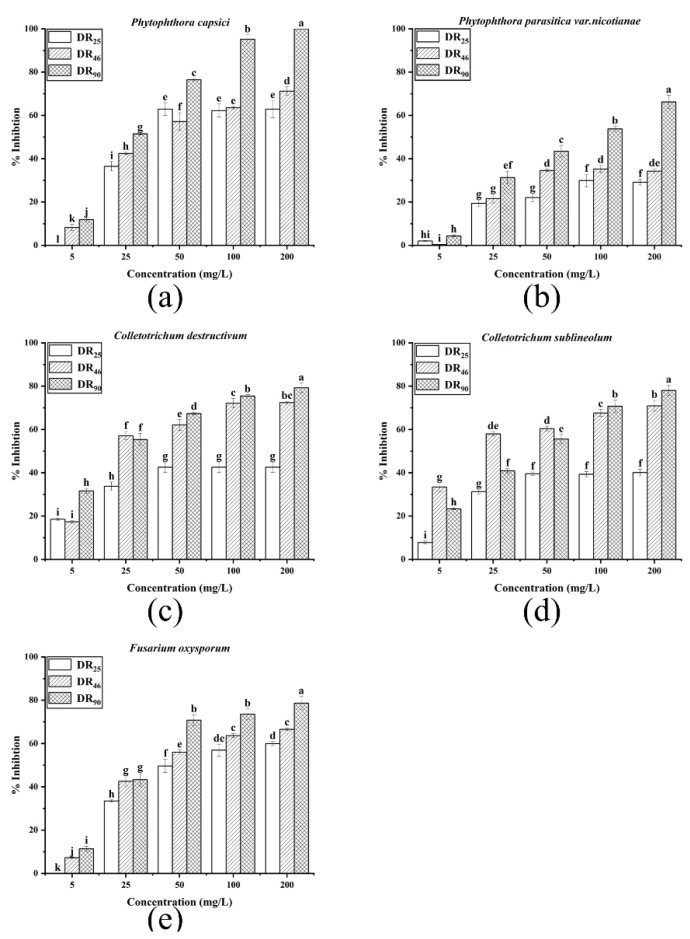
Mycelium growth inhibition rates of DR_25_, DR_46_ and DR_90_ against fungi at different concentrations (5, 25, 50, 100, 200 mg/L): (**a**) *P. capsica*; (**b**) *P. nicotianae*; (**c**) *C. destructivum*; (**d**) *C. sublineolum*; (**e**) *F. oxysporum*. Error bars represent standard deviation (SD), and letters indicate significant difference at α = 0.05 according to LSD.

**Figure 5 molecules-27-07746-f005:**
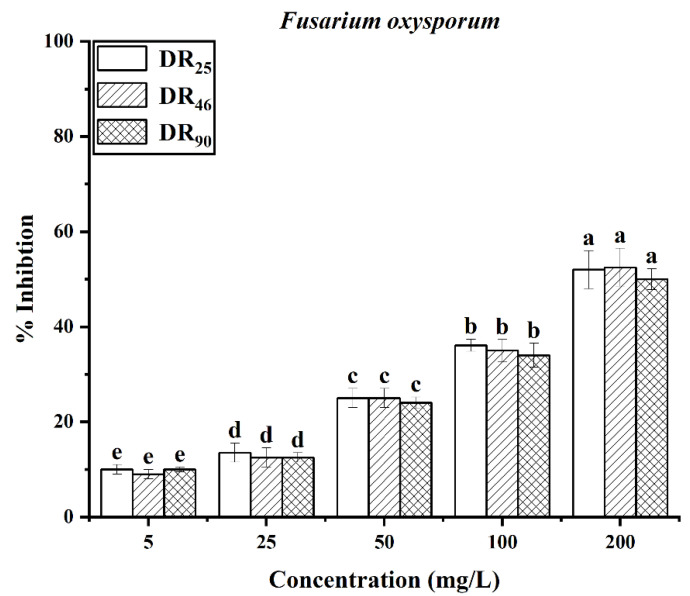
The inhibition rate of rhamnolipids against spores of *F. oxysporum*. Error bars represent standard deviation (SD), and letters indicate significant difference at α = 0.05 according to LSD.

**Table 1 molecules-27-07746-t001:** The components and purity of rhamnolipid products.

Strain	Rhamnolipids	Purity %	Homolog Ratio %		
			Rha_2_-C_10_-C_10_	Rha-C_10_-C_10_	Others
PAO1	DR_25_	60.31	25.45	66.78	7.77
HN	DR_46_	65.39	46.46	43.86	9.68
YM4	DR_90_	79.65	89.52	5.96	4.52

**Table 2 molecules-27-07746-t002:** Emulsification results of rhamnolipid products and chemical emulsifiers (SDS, EL-40) ^a^.

Emulsifier	Solvents		
	S-200	Xylene	Cyclohexanone
DR_25_	++	++	-
DR_46_	++	++	-
DR_90_	++	++	-
EL-40	+++	+++	-
SDS	++	++	+++

^a^ The rhamnolipids solutions were used at the concentration of 10 g/L, pH 7.0. Note: Emulsification effect evaluation. - No emulsification ability; + Unstable; ++ Emulsification stable; +++ Emulsification fine and stable.

**Table 3 molecules-27-07746-t003:** Emulsifying activity of DR_25_, DR_46_ and DR_90_ at various pH ^a^.

pH Value	S-200			Xylene			Cyclohexanone		
	DR_25_	DR_46_	DR_90_	DR_25_	DR_46_	DR_90_	DR_25_	DR_46_	DR_90_
5.5	+	+	+	+	+	+	-	-	-
7.0	++	++	++	++	++	++	-	-	-
8.5	+++	+++	+++	+++	+++	+++	-	-	-

^a^ The rhamnolipids solutions were used at the concentration of 10 g/L. Note: Emulsification effect evaluation. - No emulsification ability; + Unstable; ++ Emulsification stable; +++ Emulsification fine and stable.

**Table 4 molecules-27-07746-t004:** The average mycelium inhibition rates (%) of rhamnolipids (200 mg/L) against five plant pathogenic fungi.

Fungi	Control	Inhibition Rate %		
		DR_25_	DR_46_	DR_90_
*P. capsici*	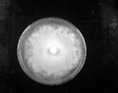	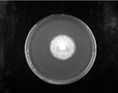 62.89 ± 4.01	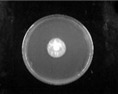 71.22 ± 1.98	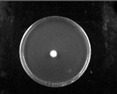 100
*P. nicotianae*	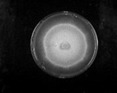	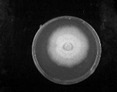 29.10 ± 1.31	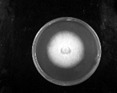 34.26 ± 1.10	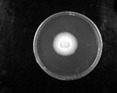 66.21 ± 3.06
*C.sublineolum*	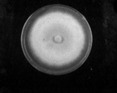	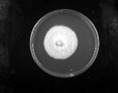 42.59 ± 2.43	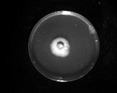 72.42 ± 0.53	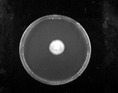 79.27 ± 2.15
*F. oxysporum*	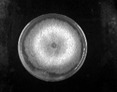	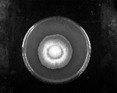 40.07 ± 1.54	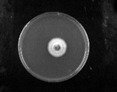 70.94 ± 2.47	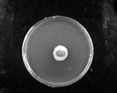 78.03 ± 2.40
*C. destructivum.*	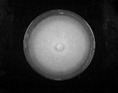	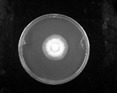 59.97 ± 0.95	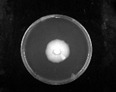 66.57 ± 0.62	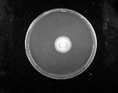 78.68 ± 3.06

## Data Availability

Not applicable.

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
