# Peer review of "Emulsifying Properties of Rhamnolipids and Their In Vitro Antifungal Activity against Plant Pathogenic Fungi"

_molecules, 2022, doi:10.3390/molecules27227746_

Round 1

Reviewer 1 Report

Accept in present form

Author Response

Thank you for your review export.

Reviewer 2 Report

Paper entitled “Emulsifying properties of rhamnolipids and their in vitro antifungal activity against plant pathogenic fungi” meets the necessary standards for publication in this journal.

Please check the entire manuscript carefully for eventual typographical errors.

 Attention when writing references. They are not unitary.

Final Conclusion: The paper meets the necessary standards for publication.

Author Response

Thank you for your advice. We had carefully examined the manuscript again

Reviewer 3 Report

Dear Colleague,

I have carefully reviewed the manuscript “Emulsifying properties of rhamnolipids and their in vitro antifungal activity against plant pathogenic fungi, " submitted to Journal molecules.

The manuscript is well written. However, I feel that further experimental work is necessary before it can be considered for publication. The authors merely described that three different kinds of rhamnolipids were extracted from laboratory fermentation. They never mentioned the identity and taxonomy of the source. It is highly recommended that the rhamnolipids-producing microorganism (isolate) should be characterized based on the 16S rRNA gene similarityIn this way, molecular and physical characterization of the organisms producing the compounds was not given.

Furthermore, nothing has been mentioned regarding the isolation procedure, characterization of the molecules, purification and homolog ratios determination. Data from analytical methods (e.g. LC-MS) used should be supplied at least in the form of supplementary material. The details mentioned above should accompany the reported bioassay results of the metabolites.  

 In addition, the antifungal and emulsifying properties of rhamnolipids have been well documented in the literature. Therefore, the quality of this manuscript could be extended if the authors provide a detailed account of the scientific contribution of the manuscript to the field.

Author Response

  1. The authors merely described that three different kinds of rhamnolipids were extracted from laboratory fermentation. They never mentioned the identity and taxonomy of the source. It is highly recommended that the rhamnolipids-producing microorganism (isolate) should be characterized based on the 16S rRNA gene similarity. In this way, molecular and physical characterization of the organisms producing the compounds was not given. Furthermore, nothing has been mentioned regarding the isolation procedure, characterization of the molecules, purification and homolog ratios determination. Data from analytical methods (e.g. LC-MS) used should be supplied at least in the form of supplementary material. The details mentioned above should accompany the reported bioassay results of the metabolites.

Thank you very much for your professional suggestion, and your advice helps to improve our paper. The rhamnolipid production strains and fermentation process were detailed in the revised manuscript (Line 63-78). The extraction, purification and characterization of rhamnolipid products were also supplemented (Line 79-82).

To better show the difference between the thee rhamnolipid products, we provided the HPLC profiles (Figure 1) to help readers understand.

  1. The antifungal and emulsifying properties of rhamnolipids have been well documented in the literature. Therefore, the quality of this manuscript could be extended if the authors provide a detailed account of the scientific contribution of the manuscript to the field.

Thanks a lot for your suggestions. We highlight the innovation and scientific contributions of the paper in the discussion section (marked in red).

In previous studies, the results make it widely agreed that homolog ratios in rhamnolipids have an important impact on performance. However, when rhamnolipids were used as pesticide emulsifier, their working concentration were usually more than 20g/L. Our results proved that the emulsification capability of rhamnolipids with various homolog ratios had little difference at high concentration. This suggests that homolog ratios of rhamnolipids have no impact on the emulsify function in pesticide. This result helps to promote the application of diverse rhamnolipids.

Round 2

Reviewer 3 Report

N/A